**Title**: What constitutes understanding of ventral pathway function?

**Scientific question**: How can we use computational models to decipher and/or represent neuronal tuning properties, and standardize descriptions for comparisons across stimuli/tasks/species?

**Introduction.** At the onset of visual neuroscience, first there was light, and then came explanations. Working to stimulate neurons in primary visual cortex (V1) in 1958, Hubel and Wiesel projected white light onto cat retinas using a modified ophthalmoscope and a slide projector. They had glass- and brass slides with drawings and cutouts, using them to shape light into simple geometric patterns. Among their many findings, they established V1 neurons showed higher activity to specifically placed line segments – lines optimized in their location, length/width, color, and their rotation. The simplicity of these stimuli allowed for straightforward interpretations, specifically that V1 neurons signal contour orientation[1].

**Observations vs. interpretations.** There were five components that made these experiments canonical, and have been included in most subsequent studies of visual neuroscience:

1) A physical stimulus (e.g., *light patterns* on a projection screen/computer monitor).
2) A generative method for producing the physical stimuli (e.g., lines drawn manually on slides, variables for a computer graphics library, vectors in a generative adversarial network).
3) An experimenter-labeled stimulus space (e.g., *orientation, categories*) with a metric to order/cluster the physical stimuli (e.g., *angular distance, perceptual similarity*).
4) Neuronal activity associated with each stimulus (e.g., *spike rates*).
5) A potential mechanism suggesting how those tuning functions could arise from earlier inputs (e.g., spatially aligned projections from neurons in the midbrain [lateral geniculate nucleus]).

The first and fourth components are observables (we refer to these as *pixels* and *spikes*). The second and third components are fundamentally entangled with the experimenter's theories and interpretations (we refer to these as *methods* and *spaces*). The linchpin observation is that in this experimental design, the relationship between pixels and spikes is causal, but the relationship between spaces and spikes is correlational. Theoretically, there can be alternative explanations implicit in any given stimulus space which also affect neuronal activity — in an experiment, the subject's brain only has access to the physical stimuli, not to the meaning attached to it.

**One reason to avoid complicated stimuli.** This gap between pixels and stimulus spaces becomes more pronounced along the cortical visual hierarchy. The primate brain is capable of recognizing patterns that are much more complicated than short contours — it is in fact designed for that — and therefore, neurons across the occipito-temporal lobes (in the monkey brain: V2, V3, V4 and inferotemporal cortex, IT) become correspondingly more complicated in their selectivity. As receptive field size increases along the occipitotemporal pathway, neurons can be tested with an enormous set of stimulus spaces. The more complicated the stimulus, the more likely it is multiple implicit interpretations can be associated with it. For example, the temporal lobe of the primate contains neurons that show strong activity in response to images of faces. Faces are physical entities in the world and knowing how to extract information from them is an essential skill, both for sociality and survival. They have many features that may be causal in eliciting a given neuron's spikes — for example, the eyes, the cranial curvature, or even unnamed feature conjunctions (the convexity of the upper right skull *plus* the contralateral eye *plus* a tan texture). Spikes might even be elicited by extra-retinal, conceptual features such as faces' social value. Complicated stimuli require additional work to narrow the most accurate and simple explanation.

**One reason not to avoid complicated stimuli.** If the relationships of pixels-to-spikes and spaces-to-spikes are not identical, one way to align them is to generate simpler stimuli, as Hubel and Wiesel did, and to extend this approach throughout the visual hierarchy. This is a productive tactic that has revealed fundamental facts about the visual system[2]. But it is also true organisms' visual systems allow them to process information in natural conditions, with crowded scenes tangling multiple visual features — shapes, contours, curves, textures — and if we are interested in defining how the visual system works (not just what we can make it do), then it is important to determine how well findings based on simplified visual stimuli generalize to the neuron's activity in the real world. In other words, we want to explain *and* predict neuronal responses across contexts.

**CNNs as prediction machines.** The ability to predict neuronal activity has been increased by convolutional neural networks (CNNs). CNNs can be trained using photographs to classify real-world scenes into semantic categories. These CNNs produce hidden unit activations which can serve as basis sets to fit linear models of a given neuron's activity. CNNs vary in their architecture, layer operations and the features they learn to represent about the natural world, and this variability makes some CNNs more suitable for neuronal fitting than others. Some CNNs allow linear models to reach an accounted-for variance of around 60% of a neuron's responses to natural images, better than any other approach[3]. This predictive success, in addition to the full mechanistic transparency of CNNs, has led to the position that the best way to explain cortical neurons is to find their ideal surrogate CNN — one predicting the neuron's activity as close to 100% of variance as possible.

**Are CNNs explanations?** CNNs are fully transparent in their mechanisms, but not in their content — that is, not in the ways they transform specific information from unit to unit or layer to layer. We may know channel 20 of AlexNet layer conv4 performs a convolution, but what convolution exactly? What visual world attributes does that filter emphasize or minimize? There are extensive efforts in machine learning to visualize the filtering operations characterizing given hidden units, relying on gradient descent or generative models. These efforts, also paralleled in neuroscience, are evidence that at best, CNNs are only partial explanations. Having a CNN that makes perfect predictions of a given neuron's behavior is an essential goal, but not one that would settle most questions in this scientific marketplace.

**Challenge or controversy**: In machine learning, the question of what units represent and how they accomplish a given visual task remains hotly contested. On the other extreme, some investigators argue that "interpretability" is neither necessary nor desirable. However, rising societal consensus requires the use of models that can explain their own decisions[4], particularly in high-stake applications where such models could fail (e.g., providing accountability in self-driving car accidents, recognizing criminals from photos, predicting future recidivism probability; assessing qualifications of a job/loan applicant).

In parallel, Neuroscience is undergoing a major revolution. Some investigators use **word-based** descriptions to characterize neuronal tuning axes (orientation tuning, color tuning, motion-direction tuning, face cells, body part areas, Jennifer Anniston cells; this trend continues beyond vision with place cells, reward cells, error monitoring cells, and more). Sharply departing from word models, other investigators argue for quantitative descriptions based on computational models and numbers (curvature models, motion energy models, hierarchical models like HMAX, convolutional neural network models, and more).

At the heart of this controversy is the challenge of agreeing on what we need for a theory of vision. Fundamental desiderata include: understanding (hard to define but an appealing feature), falsifiability (fundamental to scientific theories), and predictability (critical to evaluate generalization and for practical applications).

**Competing hypotheses**. Here we characterize types of experimental approaches in vision studies outlined as competing (although they might be best seen as complementary) and illustrate challenges to each.

*Tuning functions on* a priori *dimensions*. These can be low-dimensional (orientation) or high-dimensional (e.g., parametric deformations of faces, 3D geometric space). A typical limitation is that the space covered by the dimensionality is incomplete.

*Tuning functions on* a posteriori *dimensions*. This is a typical result from PCA and related analyses. Some claim that the ordering of exemplars along a PCA dimension is an explanation (this overlaps with images as explanations, see below). Sometimes subjective descriptions are applied to those images.

*Predictive models with interpretable operations*. These include typical models such as those for direction tuning[5]. These are limited in domain like tuning functions and limited in application to low-dimensional domains.

*Images.* These are the results of DeepDream[6]-like analyses of both CNNs and neurons[7–9]. They used to be the main type of result for neurophysiology experiments on vision. One could argue that the images themselves are an explanation. One can also apply subjective descriptions, though many think those are speculative and dissatisfying.

*Predictive models with potentially interpretable operations.* Based on CNNs. Some think such models are the endpoint, while others think that without extensive further analyses, CNNs that replicate function are no more an explanation than the brain itself is.

*Causal manipulations.* Some think that all correlative evidence (above) is inconclusive, and only causal experiments advance understanding. Unfortunately, the dimensionality of causal manipulations so far is very low compared to the dimensionality of object vision.

The challenge would be to achieve a common experimental framework that overcomes these existing weaknesses of dimensionality and comprehensibility.

**Concrete outcomes:**

- To gather teams working in mid-to-anterior visual cortex, and to outline every team's desiderata along descriptions of neuronal selectivity.

- Develop a way to track progress in the field's collective ability to compare, falsify and stress-test the competing approaches by testing them all on the same cortical neurons. For example, using a set S1 of visual stimuli, each team would collect responses and develop a predictive model, then present a set S2 of stimuli with different content and visual statistics. The challenge would be to predict the neuronal activity, but also explain the inner workings of the model in terms of both mechanism and content transformations.

- To have more comprehensive dissections of CNN-models fitted to neural data, including visualizations (https://openai.com/blog/microscope/)

- To encourage research teams to actively include hypotheses in their paper discussions on how any given stimulus space could arise in the brain.

- To develop shared code repositories across labs, stimulus datasets, neuronal data, and a minimal set of guidelines on mechanisms to falsify theories.

**Benefit to the community:** Although we focus on vision as a paradigmatic example, the same questions apply to all sensory and perceptual neuroscience. Historically, vision has led the way in many questions. We expect that the results, data, code, mechanisms advanced here will spread to the community. The conclusions reached through this work have the potential to radically transform how we do neuroscience, by developing an infrastructure of code, data and ideas available to every lab. The framing of electrophysiology-based explanations through computational models provides a natural path to machine learning and intelligence applications, including improvements to computer vision algorithms, neural networks that better match primate perception, and potential future ways to stimulate cortex for prosthetic applications.

**Initial organizers:** Charles Connor (JHU), Gabriel Kreiman (HMS), Carlos R. Ponce (WUSTL). **Contributors:** Carl Craver (WUSTL), Margaret Livingstone (HMS), Martin Schrimpf (MIT), Binxu Wang (WUSTL). **Commitments:** If chosen as a topic, members will work to incorporate feedback from the community and invite particular CCN members to the GAC. Together, these members will run the kickoff workshop and write a position paper. Updates will be presented at the following CCN2022.

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
