# OpenReview forum: "What constitutes understanding of ventral pathway function?"
_ccneuro.org/CCN/2021/Workshop/GAC_

### Official Review · ~Adrien_Doerig1 · 2021-07-23
**Towards a consensus?**

**Rating:** 7
**Confidence:** 4

**Review:**

Connor et al. propose to address the important and difficult question of what would constitute an understanding of visual processing. As they lay out in their proposal, there are currently many approaches to do so, each with strengths and weaknesses. The authors propose to gather researchers into a coherently organized research effort, with well defined hypotheses for each group, harmonized methodologies and shared code, datasets and results.

A field-wide consensus about what we are trying to explain would obviously be extremely beneficial. For this reason, I support this proposal.

One limitation is that this proposal does not seem to fit exactly in the generative adversarial collaboration, since there are not two well defined camps that propose to adversarially collaborate to resolve a question using a well-defined approach. Instead, it seems more like an exploratory collaboration. There is a slight concern that this may make the collaboration less adversarial and therefore a little off topic for this "venue", and perhaps a little less effective. However, I am not very familiar with this GAC format yet, so I may be wrong.